# What Are the Invariant Occlusive Components of Image Patches? A Probabilistic Generative Approach

**Zhenwen Dai**
University of Sheffield, UK, and
FIAS, Goethe-University Frankfurt, Germany
z.dai@sheffield.ac.uk

**Georgios Exarchakis**
Redwood Center for Theoretical Neuroscience,
The University of California, Berkeley, US
exarchakis@berkeley.edu

**Jörg Lücke**
Cluster of Excellence Hearing4all, University of Oldenburg, Germany,
and BCCN Berlin, Technical University Berlin, Germany
joerg.luecke@uni-oldenburg.de

## Abstract

We study optimal image encoding based on a generative approach with non-linear feature combinations and explicit position encoding. By far most approaches to unsupervised learning of visual features, such as sparse coding or ICA, account for translations by representing the same features at different positions. Some earlier models used a separate encoding of features and their positions to facilitate invariant data encoding and recognition. All probabilistic generative models with explicit position encoding have so far assumed a linear superposition of components to encode image patches. Here, we for the first time apply a model with non-linear feature superposition and explicit position encoding for patches. By avoiding linear superpositions, the studied model represents a closer match to component occlusions which are ubiquitous in natural images. In order to account for occlusions, the non-linear model encodes patches qualitatively very different from linear models by using component representations separated into mask and feature parameters. We first investigated encodings learned by the model using artificial data with mutually occluding components. We find that the model extracts the components, and that it can correctly identify the occlusive components with the hidden variables of the model. On natural image patches, the model learns component masks and features for typical image components. By using reverse correlation, we estimate the receptive fields associated with the model's hidden units. We find many Gabor-like or globular receptive fields as well as fields sensitive to more complex structures. Our results show that probabilistic models that capture occlusions and invariances can be trained efficiently on image patches, and that the resulting encoding represents an alternative model for the neural encoding of images in the primary visual cortex.

## 1 Introduction

Probabilistic generative models are used to mathematically formulate the generation process of observed data. Based on a good probabilistic model of the data, we can infer the processes that have generated a given data point, i.e., we can estimate the hidden causes of the generation. These hidden causes are usually the objects we want to infer knowledge about, be it for medical data, biological processes, or sensory data such as acoustic or visual data. However, real data are usually very complex, which makes the formulation of an exact data model infeasible. Image data are a typical example of such complex data. The true generation process of images involves, for instance, different objects with different features at different positions, mutual occlusions, object shades, lighting

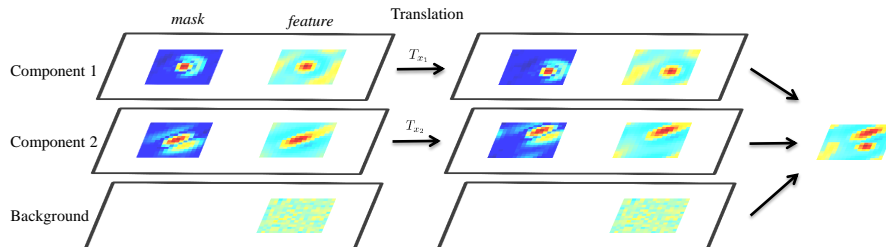

Figure 1: An illustration of the generation process of our model.

conditions and reflections due to self-structure and nearby objects. Even if a generative model can capture some of these features, an inversion of the model using Bayes' rule very rapidly becomes analytically and computationally intractable. As a consequence, generative modelers make compromises to allow for trainability and applicability of their generative approaches.

Two properties that have, since long, been identified as crucial for models of images are object occlusions [1–5] and the invariance of object identity to translations [6–13]. However, models incorporating both occlusions and invariances suffer from a very pronounced combinatorial complexity. They could, so far, only be trained with very low dimensional hidden spaces [2, 14, 15]. At first glance, occlusion modeling is, furthermore, mathematically more inconvenient. For these reasons, many studies including style and content models [16], other bi-linear models [17, 18], invariant sparse coding [19, 20], or invariant NMF [21] do not model occlusions. Analytical and computation reasons are often explicitly stated as the main motivation for the use of the linear superposition of components (see, e.g., [16, 17]).

In this work, we for the first time study the encoding of natural image patches using a model with both non-linear feature combinations and translation invariances.

## 2   A Generative Model with Non-linear and Invariant Components

The model used to study image patch encoding assumes an *exclusive* component combination, i.e., for each pixel exclusively one cause is made responsible. It thus shares the property of exclusiveness with visual occlusions. The model will later be shown to capture occluding components. We will, however, not model explicit occlusion using a depth variable (compare [2]) but will focus on the exclusiveness property. The applied model is a novel version of the *invariant occlusive components* model studied for mid-level vision earlier [22]. We first briefly reiterate the basic model in the following and discuss the main differences of the new version afterwards.

We consider image patches $\vec{y}$ with $D^2$ observed scalar variables, $\vec{y} = (y_1, \ldots, y_{D^2})$. An image patch is assumed to contain a subset from a set of $H$ components. Each component $h$ can be located at a different position denoted by an index variable $x_h \in \{1, \ldots, D^2\}$, which is associated with a set of permutation matrices that covers all the possible planar translations $\{T_1, \ldots, T_{D^2}\}$ (similar formulations have also been used in sprite models [14, 15]). Each component $h$ is modeled to appear in an image patch with probability $\pi_h \in (0, 1)$. Following [22], we do not model component presence and absence explicitly but, for mathematical convenience, assign the special 'position' $-1$ to all the components which are not chosen to generate the patch. Assuming a uniform distribution for the positions, the prior distribution for components and their positions is thus given by:

$$p(\vec{x}|\vec{\pi}) = \prod_h p(x_h|\pi_h), \ p(x_h|\pi_h) = \begin{cases} 1 - \pi_h, & x_h = -1 \\ \frac{\pi_h}{D^2}, & \text{otherwise} \end{cases}, \tag{1}$$

where the hidden variable $\vec{x} = (x_1, \ldots, x_H)$ contains the information on presence/absence and position of all the image components.

In contrast to linear models, the studied approach requires two sets of parameters for the encoding of image components: component *masks* and component *features*. Component masks describe *where* an image component is located, and component features describe *what* a component encodes (compare [2, 3, 14, 15]). High values of mask parameters $\vec{\alpha}_h$ encode the pixels most associated with a component $h$ but the encoding has to be understood relative to a global component position. The feature parameters $\vec{w}_h$ encode the values of a component's features. Fig. 1 shows an example

of the mask and feature parameters for two typical low-level visual features. Given a particular position, the mask and feature parameters of the component are transformed to the target position by multiplying a corresponding translation matrix like $T_{x_h} \vec{\alpha}_h$ and $T_{x_h} \vec{w}_h$. When generating an image patch, two or more components may occupy the same pixel, but according to occlusion the pixel value is exclusively determined by only one of them. This exclusiveness is formulated by defining a *mask* variable $\vec{m} = (m_1, \ldots, m_{D^2})$. For a pixel at a position $d$, $m_d$ determines which component is responsible for the pixel value. Therefore, $m_d$ takes a value from the set of present components $\Gamma = \{h | x_h \neq -1\}$ plus a special value "0" indicating background, and the prior distribution of $\vec{m}$ is defined as:

$$p(\vec{m}|\vec{x}, A) = \prod_{d=1}^{D^2} p(m_d|\vec{x}, A), \quad p(m_d = h|\vec{x}, A) = \begin{cases} \frac{\alpha_0}{\alpha_0 + \sum_{h' \in \Gamma} (T_{x_{h'}} \vec{\alpha}_{h'})_d}, & h = 0 \\ \frac{(T_{x_h} \vec{\alpha}_h)_d}{\alpha_0 + \sum_{h' \in \Gamma} (T_{x_{h'}} \vec{\alpha}_{h'})_d}, & h \in \Gamma \end{cases}, \tag{2}$$

where $A = (\vec{\alpha}_1, \ldots, \vec{\alpha}_H)$ contains the mask parameters for all the components, and $\alpha_0$ defines the mask parameter for background. The mask variable $m_d$ chooses the component $h$ with a high likelihood if the translated mask parameter of the corresponding component is high at the position $d$. Note that $m_d$ follows a mixture model given the presence/absence and positions of all the components $\vec{x}$. This can be thought of as an approximation to the distribution of mask variables marginalizing the depth orderings and pixel transparency in the exact occlusive model (see Supplement A for a comparison). After drawing the values of the hidden variables $\vec{x}$ and $\vec{m}$, an image patch can be generated with a Gaussian noise model, which is given by:

$$p(\vec{y}|\vec{m}, \vec{x}, \Theta) = \prod_{d=1}^{D^2} p(y_d|m_d, \vec{x}, \Theta), \quad p(y_d|m_d = h, \vec{x}, \Theta) = \begin{cases} \mathcal{N}(y_d; B, \sigma_B^2), & h = 0 \\ \mathcal{N}(y_d; (T_{x_h} \vec{w}_h)_d, \sigma^2), & h \in \Gamma \end{cases}, \tag{3}$$

where $\sigma^2$ is the variance of components, and $\Theta = (\vec{\pi}, W, A, \sigma^2, \alpha_0, B, \sigma_B^2)$ are all the model parameters. The background distribution is a Gaussian distribution with mean $B$ and variance $\sigma_B^2$. Compared to an occlusive model with exact EM (see Supplement A), our approach will use the exclusiveness approximation and a truncated posterior approximation in order to make learning tractable.

The model described in (1) to (3) has been optimized for the encoding of image patches. First, feature variables are scalar to encode light intensities or input by the lateral geniculus nucleus (LGN) rather than color features for mid-level vision. Second, to capture the frequency of presence for individual components, we implement the learning of the prior parameter of presence $\vec{\pi}$. Third, the pre-selection function in the variational approximation (see below) has been adapted to the usage of scalar valued features. As a scalar value is much less distinctive than the sophisticated image features used in [22], the pre-selection of components has been changed to the complete component instead of only salient features.

## 3 Efficient Likelihood Optimization

Given a set of image patches $Y = (\vec{y}^{(1)}, \ldots, \vec{y}^{(n)})$, learning is formulated as inferring the best model parameters w.r.t. the log-likelihood $L = p(Y|\Theta)$. Following the Expectation Maximization (EM) approach, the parameter update equations are derived. The equations of the mask parameter $\vec{\alpha}_h$, and feature parameter $\vec{w}_h$ are the same as in [22]. Additionally, we derived the update equation for the prior parameter of presence:

$$\pi_h = \frac{1}{N} \sum_{n=1}^{N} \sum_{\vec{x} \in \{x_h \neq -1\}} p(\vec{x}|\vec{y}^{(n)}, \Theta). \tag{4}$$

By learning the prior parameters $\pi_h$, the probabilities of individual components' presence can be estimated. This allows us to gain more insights about the statistics of image components. In the update equations, a posterior distribution has been estimated for each data point, which corresponds to the E-step of an EM algorithm. The posterior distribution of our model can be decomposed as:

$$p(\vec{m}, \vec{x}|\vec{y}, \Theta) = p(\vec{x}|\vec{y}, \Theta) \prod_{d=1}^{D^2} p(m_d|\vec{x}, \vec{y}, \Theta), \tag{5}$$

in which $p(\vec{x}|\vec{y}, \Theta)$ and $p(m_d|\vec{x}, \vec{y}, \Theta)$ are estimated separately. Computing the exact distribution of $p(\vec{x}|\vec{y}, \Theta)$ is intractable, as it includes the combinatorics of the presence/absence of components and their positions. An efficient posterior approximation, Expectation Truncation (ET), has been successfully employed. ET approximates the posterior distribution as a truncated distribution [23]:

$$p(\vec{x}|\vec{y}, \Theta) \approx \frac{p(\vec{y}, \vec{x}|\Theta)}{\sum_{\vec{x}' \in \mathcal{K}_n} p(\vec{y}, \vec{x}'|\Theta)}, \text{ if } \vec{x} \in \mathcal{K}_n, \tag{6}$$

and zero otherwise. If $\mathcal{K}_n$ is chosen to be small but to contain the states with most posterior probability mass, the computation of the posterior distribution becomes tractable while a high accuracy

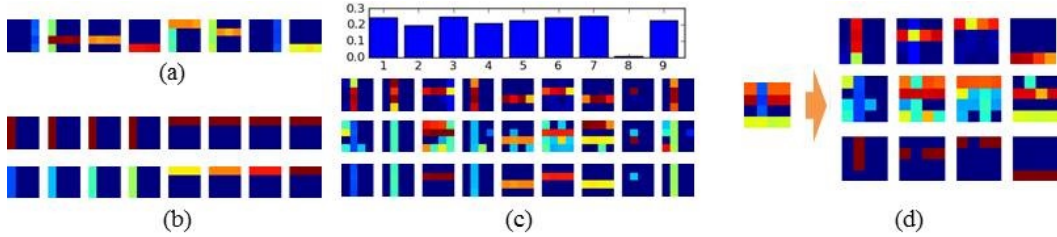

Figure 2: Numerical Experiments on Artificial Data. (a) eight samples of the generated data sets. (b) The parameters of the eight components used to generate the data set. The 1st row contains the binary transparency parameters and the 2nd row shows the feature parameters. (c) The learned model parameters ($H = 9$). The top plot shows the learned prior probabilities $\vec{\pi}$. The 1st row shows the mask parameters $A$; the 2nd row shows the feature parameters $W$; the 3rd row gives a good visualization of only the frequent used elements/pixels (setting the feature parameter $w_{hd}$ of the elements/pixels with $\alpha_{hd} < 0.5$ to zero). (d) The result of inference given a image patch (shown on the left). The right side shows the four components inferred to be present (each takes a column). The 1st and 2nd rows show the mask and features parameters shifted according to the MAP inference $\vec{x}^{\text{MAP}}$, and the 3rd row shows the inferred posterior $p(m_d|\vec{x}^{\text{MAP}}, \vec{y}, \Theta)$. All the plots are heat map (Jet color map) visualizations of scalar values.

of the approximations can be maintained [23]. To select a proper subspace $\mathcal{K}_n$, $\tau$ features (pixel intensities) are chosen according to their mask parameters. Based on the chosen features, a score value $\mathcal{S}(x_h)$ is computed for each component at each position (see [22]). We select $H'$ components, denoted as $\mathcal{H}$, for the candidates that may appear in the given image according to the probability $p(\vec{y}, \tilde{x}_h|\Theta)$. $\tilde{x}_h$ corresponds to the vector $\vec{x}$ with $x_h = x_h^*$ and the rest components absent ($x_{h'} = -1, h' \neq h$), where $x_h^*$ is the best position of the component $h$ w.r.t. $\mathcal{S}(x_h)$. This is different from the earlier work [22], where $\mathcal{K}_n$ is constructed directly according to $\mathcal{S}(x_h)$. For each component, we select the set of its candidate positions $\mathcal{X}_h$, $x_h \in \mathcal{X}_h$, which contains the $p$ best positions w.r.t. $\mathcal{S}(x_h)$. Then the truncated subspace $\mathcal{K}_n$ is defined as:

$$\mathcal{K}_n = \{\vec{x} \,|\, (\sum_j s_j \leq \gamma \text{ and } s_i = 0, \forall i \notin \mathcal{H}) \text{ or } \sum_{j'} s_{j'} \leq 1\}, \tag{7}$$

where $s_h$ represents the presence/absence state of the component $h$ ($s_h = 0$ if $x_h = -1 \cup x_h \notin \mathcal{X}_h$ and $s_h = 1$ if $x_h \in \mathcal{X}_h$). To avoid converging to local optima, we used the *directional annealing* scheme [22] for our learning algorithm.

## 4   Numerical Experiments on Artificial Data

The goal of the experiment on artificial data is to verify that the model and inference method can recover the correct parameters, and to investigate inference on the data generated according to occlusions with explicit depth variable. We generated $4 \times 4$ gray-scale image patches. In the data set, eight different components are used, which are four vertical 'bars' and four horizontal 'bars', and each bar has a different intensity and has a binary vector indicating its 'transparency' (1 for non-transparent and 0 for transparent, see Fig. 2b) . When generating an image patch, a subset of components is selected according to their prior probabilities $\pi_h = 0.25$, and the selected components are combined according to a random depth ordering (flat priors on the ordering). A component with smaller depth will occlude the components with larger depth, and for each image patch we sample a new depth-ordering. For the pixels in which all the selected components are transparent, the value is determined according to the background with zero intensity ($B = 0$). All the pixels generated by components are subject to a Gaussian noise with $\sigma = 0.02$ and the pixels belonging to the background have a Gaussian noise with $\sigma_B = 0.001$. In total, we generated $N = 1,000$ image patches. Fig. 2a shows eight samples. The artificial data is similar to data generated by the *occlusive components analysis* model (OCA; [2]), except of the use of scalar features and the assumption of shift-invariance.

Fig. 2c shows the learned model parameters on the generated data set. We learned nine components ($H = 9$). The initial feature value $W$ was set to randomly selected data points. The initial mask parameter $A$ was independently and uniformly drawn from the interval $(0, 1)$. The initial annealing temperature was set to $T = 5$. After keeping constant for 20 iterations, the temperature linearly decreased to 1 in 100 iterations. For the robustness of learning, $\sigma$ decreased together with the temperature from 0.2 to 0.02, and an additive Gaussian noise with zero mean and $\sigma_w = 0.04$ was

injected into $W$ and $\sigma_w$ gradually decreased to zero. The algorithm terminated when the temperature was equal to 1 and the difference of the pseudo data log-likelihood of two consecutive iterations was sufficiently small (less than 0.1%). The approximation parameters used in learning was $H' = 8$, $\gamma = 4$, $p = 2$ and $\tau = 3$. In this result, all the eight generative components have been successfully learned. The 2nd to last component (see Fig. 2c) is a dumpy component (low $\pi_h$, i.e., very rarely used). Its single pixel structure is therefore an artifact. With the learned parameters, the model could infer the present components, their positions and the pixel-to-component assignment. Fig. 2d shows a typical example. Given an image patch on the left, the present components and their positions are correctly inferred. Furthermore, as shown on the 3rd row, the posterior probabilities of the mask variable $p(m_d|\vec{x}, \vec{y}, \Theta)$ give a clear assignment of the contributing component for each pixel. This information is potentially very valuable for tasks like parts-based object segmentation or to infer the depth ordering among the components. We assess the reliability of our learning algorithm by repeating the learning procedure with the same configuration but different random parameter initializations. The algorithm recovers all the generative components in 11 out of 20 repetitive runs. The 9 runs not recovering all bars did still recover reasonable solutions with usually 7 bars out of 8 bars represented. In general, optima of bar stimuli seem to have much more pronounced local optima, e.g., compared to image patches.

## 5 Numerical Experiments on Image Patches

After we verified the inference and learning algorithm on artificial data, it was applied to patches of natural images. As training set we used $N = 100,000$ patches of size $16 \times 16$ pixels extracted at random positions from random images of the van Hateren natural image database [24]. We modeled the sensitivity of neurons in the LGN using a difference-of-Gaussians (DoG) filter for different positions, i.e., we processed all patches by convolving them with a DoG kernel. Following earlier studies (see [5] for references), the ratio between the standard deviation of the positive and the negative Gaussian was chosen to be $1/3$ and the amplitudes chosen to obtain a mean-free center-surround filter. Fig. 3a shows some samples of the image patches after preprocessing.

Our algorithm learned $H = 100$ components from the natural image data set. The model parameters were initialized in the same way as for artificial data. The annealing temperature was initialized with $T = 10$, kept constant for 10 iterations, the temperature linearly decreased to 1 in 100 iterations. $\sigma$ decreased together with the temperature from 0.5 to 0.2, and an additive Gaussian noise with zero mean and $\sigma_w = 0.2$ was injected into $W$ and $\sigma_w$ gradually decreased to zero. The approximation parameters used for learning were $H' = 6$, $\gamma = 4$, $p = 2$ and $\tau = 50$. After 134 iterations, the model parameters had essentially converged.

Figs. 3bc show the learned mask parameters and the learned feature values for all the 100 components. Mask parameters define the frequently used areas within a component, and feature parameters reveal the appearance of a component on image patches. As can be observed, image components are very differently represented from linear models. See the component in Fig. 3d as an example: mask parameters are localized and all positive; feature parameters have positive and negative values across the whole patch. Masks and features can be combined to resemble a familiar Gabor function via point-wise multiplication (see Fig. 3d). All the above shown component representations are sorted in descending order according to the learned prior probabilities of occurrence $\vec{\pi}$ (see Fig. 3e).

## 6 Estimation of Receptive Fields

For visualization, mask and feature parameters can be combined via point-wise multiplication. To more systematically and quantitatively interpret the learned components and to compare them to biological experimental findings, we estimated the predicted receptive fields (RFs). RFs estimates were computed with reverse correlation based on the model inference results. Reverse correlation can be defined as procedure to find the best linear approximation of the components' presence given an image patch $\vec{y}^{(n)}$. More formally, we search for a set of predicted receptive fields $\vec{R}_h, h \in \{1, \ldots, H\}$ that minimize the following cost function:

$$f = \frac{1}{N} \sum_n \sum_{\vec{x} \in \mathcal{K}_n} p(\vec{x}|\vec{y}^{(n)}, \Theta) \sum_h (\vec{R}_h^T \bar{T}_{x_h} \vec{y}^{(n)} - s_h)^2 + \lambda \sum_h \vec{R}_h^T \vec{R}_h, \qquad (8)$$

where $\vec{y}^{(n)}$ is the $n$th stimulus and $\lambda$ is the coefficient for L2 regularization. $s_h$ is a binary variable representing the presence/absence state of the component $h$, where $s_h = 0$ if $x_h = -1$, and $s_h = 1$

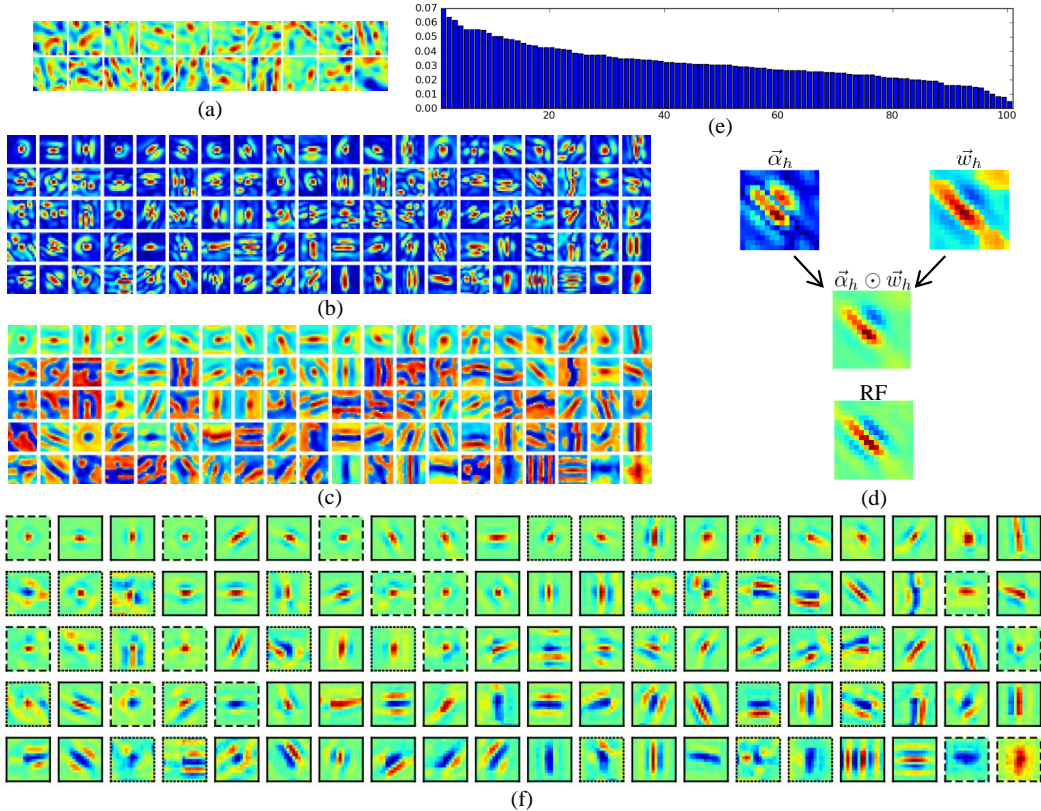

Figure 3: The invariant occlusive components from natural image patches. (a) shows 20 samples of the pre-processed image patches. (b) shows the mask parameter and (c) shows the feature parameter. (d) shows an example of the relation with the learned model parameters and the estimated RFs. (e) shows the learned prior probabilities $\vec{\pi}$. (f) shows the estimated Receptive Fields (RF). The RFs were fitted with 2 dimensional Gabor and DoG functions. The dashed line marks the RFs that have a more globular structure. The solid lines mark the RFs the were fitted accurately by a Gabor function. The dotted lines marks the RFs that were not approximated very well by the fitted function. All the shown model parameters in (b-c) and receptive fields in (f) are sorted in descent according to $\vec{\pi}$. The plots (a-d) and (f) are heat map visualization with local scaling on individual fields (Jet color map), and (a), (c) and (f) fix light green to be zero.

otherwise. As our model allows the components to be at different locations, the reverse correlation is computed by shifting the stimuli according to the inferred location of each components. $\bar{T}_{x_h}$ represents the transformation matrix applied to the stimulus for the component $h$, which is the opposite transformation of the inferred transformation $T_{x_h}$ ($\bar{T}_{x_h} T_{x_h} = \mathbb{1}$). For the absent components, the stimulus is used without any transformations ($T_{-1} = \mathbb{1}$).

Due to the intractability of computing an exact posterior distribution, given a data point, the cost function only sums across the truncated subspace $\mathcal{K}_n$ in the variational approximation (see Sec. 3). By setting the derivative of the cost function to zero, $\vec{R}_h$ can be estimated as:

$$\vec{R}_h = \left(\lambda N \mathbb{1} + \sum_n \langle \bar{T}_{x_h} \vec{y}^{(n)} (\bar{T}_{x_h} \vec{y}^{(n)})^T \rangle_{q_n}\right)^{-1} \left(\sum_n \langle \vec{s}(\bar{T}_{x_h} \vec{y}^{(n)})^T \rangle_{q_n}\right) \qquad (9)$$

where $\langle \cdot \rangle_{q_n}$ denotes the expectation value w.r.t. the posterior distribution $p(\vec{x}\,|\vec{y}^{(n)}, \Theta)$ and $\mathbb{1}$ is an identity matrix. When solving $\vec{R}_h$, we often observe that many of the eigenvalues of the data covariance matrix $\sum_{n=1}^{N} \langle \bar{T}_{x_h} \vec{y}^{(n)} (\bar{T}_{x_h} \vec{y}^{(n)})^T \rangle_{q_n}$ are close to zero, which makes the solution of $\vec{R}_h$ very unstable. Therefore, we introduce a L2 regularization to the cost function. The regularization coefficient $\lambda$ is chosen between the minimum and maximum element of the data covariance matrix. The estimated receptive fields are not sensitive to the value of the regularization coefficient $\lambda$ as long as $\lambda$ is large enough to resolve the numerical instability (see Supplement for a comparison of the receptive fields estimated with different $\lambda$ values). From the experiments with artificial data and

natural image patches, we observed that the L2 regularization successfully eliminated the numerical stability problem.

Fig. 3f shows the RFs estimated according to our model. For further analysis, we matched the RFs using Gabor functions and DoG functions as was suggested in [5]. If we factored in the occurrence probabilities, we found that the model considered about 17% of all components of the patches to be globular, 56% to be Gabor-like and 27% to have another structure (see Supplement for details). The prevalence of 'center-on' globular fields may be a consequence of the prevalence of convex object shapes.

## 7  Discussion

The encoding of image patches investigated in this study separates feature and position information of visual components. Functionally, such an encoding has been found very useful, e.g., for the construction of object recognition systems. Many state-of-the-art systems for visual object classification make use of convolutional neural networks [12, 25, 26]. Such networks compute the responses of a set of filters for all positions in a predefined area and use the maximal response for further processing ([12] for a review). If we identify the predefined area with one image patch as processed by our approach, then the encoding studied here is to some extent similar to convolutional networks: (A) it uses like convolutional networks one set of component parameters for all positions; and (B) a hidden component variable of the generative model integrates or '*pools*' the information across all positions. As the here studied approach is based on a generative data model, the integration across positions can directly be interpreted as inversion of the generation process. Crucially, the inversion can take occlusions of visual features into account while convolutional networks do not model occlusions. Furthermore, the generative model uses a probabilistic encoding, i.e., it assigns probabilities to positions and features of a joint feature and position space. Ambiguous visual input can therefore be represented appropriately. In contrast, convolutional networks use one position for each feature as representation. In this sense a convolutional encoding could be regarded as MAP estimate for the feature position while the generative integration could be interpreted as *probabilistic pooling*. Many bilinear models have also been applied to image patches, e.g., [17, 18]. Such studies do report that neurally plausible receptive fields (RFs) in the form of Gabor functions emerge [17, 18]. Likewise, invariant versions of NMF [21] or ICA (in the form of ISA [9] have been applied to image patches.

In addition to Gabors, we observed in our study a large variety of further types of RFs. Gabor filters with different orientations, phase and frequencies, as well as globular fields and fields with more complex structures (Fig. 3f). Gabors have been studied since several decades, globular and more complex fields have attracted attention in the last couple of years. In particular, globular fields have attracted attention [5, 27, 28] as they have been reported together with Gabors in macaques and other species ([29] and [5] for further references). Such fields have been associated with occlusions before [5, 28, 30]; and our study now for the first time reports globular fields for an occlusive and translation invariant approach. The results may be taken as further evidence of the connection between occlusions and globular fields. However, also linear convolutional approaches have recently reported such fields [19, 31]. Linear approaches seem to require a high degree of overcompleteness or specific priors while globular fields naturally emerge for occlusion-like non-linearities. More concretely: for non-invariant linear sparse coding, globular fields only emerged from a sufficiently high degree of overcompleteness onwards [32, 33] or with specific prior settings and overcompleteness [27]; for non-invariant occlusive models [5, 30] globular fields always emerge alongside Gabors for any overcompleteness. The results reported here can be taken as confirming this observation for position invariant encoding. The invariant non-linear model assigns high degrees of occurrences (high $\pi_h$) to Gabor-like and to globular fields (first rows in Fig. 3f). Components with more complex structures are assigned lower occurrence frequencies. In total the model assumes a fraction between 10 and 20% of all data components to be globular. Such high percentages may be related to the high percentages of globular fields ($\sim$16-23%) measured in *vivo* ([29] and [5] for references). In contrast, the highest degrees of occurrences, e.g., for convolutional matching pursuit [31] seems to be assigned exclusively to Gabor features. Globular fields only emerge (alongside other non-Gabor fields) for higher degrees of overcompleteness. A direct comparison in terms of occurrence frequencies is difficult because the linear models to not infer occurrence frequencies from data. The closest match to such frequencies would be an (inverse) sparsity which is set by hand for almost all linear approaches. The reason is the use of MAP-based point-estimates while our approach uses a more probabilistic posterior estimate.

Because of their separate encoding of features and positions, all models with separate position encoding can represent high degrees of over-completeness. Convolutional matching pursuit [31] shows results for up to 64 filters of size $8 \times 8$. With 8 horizontal and 8 vertical shifts, the number of non-invariant components would amount to $8 \times 8 \times 64 = 3136$. Convolutional sparse coding [19] reports results by assuming 128 components for $9 \times 9$ patches. The number of non-invariant components would therefore amount to $10,368$. For our network we obtained results for up to 100 components of size $16 \times 16$. With 16 horizontal and 16 vertical shift this amounts to $25,600$ non-invariant components. In terms of components per observed variable, invariant models are therefore now computationally feasible in a regime the visual cortex is estimated to operate in [33].

The hidden units associated with component feature are fully translation invariant. In terms of neural encoding, their insensitivity to stimulus shifts would therefore place them into the category of V1 complex cells. Also globular fields or fields that seem sensitive to structures such as corners would warrant such units the label 'complex cell'. No hidden variable in the model can directly be associated with simple cell responses. However, a possible neural network implementation of the model is an explicit representation of component features at different positions. The *weight sharing* of the model would be lost but units with explicit non-invariant representation could correspond to simple cells. While such a correspondence can connect our predictions to experimental studies of simple cells, recently developed approaches for the estimation of translation invariant cell responses [34, 35] can represent a more direct connection. To approximately implement the non-linear generative model neurally, the integration of information would have to be a very active process. In contrast to passive pooling mechanisms across units representing linear filters (such as simple cells), it would involve neural units with explicit position encoding. Such units would control or 'gate' the information transfer from simple cells to downstream complex cells. As such our probabilistic model can be related to ideas of active *control units* for individual components [6, 7, 10, 11, 36] (also compare [37]). A notable difference to all these models is that the here studied approach allows to interpret active control as optimal inference w.r.t. a generative model of translations and occlusions.

Future work can go in different directions. Different transformations could be considered or learned [37], explicit modeling in time could be incorporated (compare [17]), and/or further hierarchical stages could be considered. The crucial challenge all such developments face are computational intractabilities due to large combinatorial hidden spaces. Base on the presented results, we believe, however, that advances in analytical and computational training technology will enable an increasingly sophisticated modeling of image patches in the future.

**Acknowledgement.**
We thank Richard E. Turner for helpful discussions and acknowledge funding by DFG grant LU 1196/4-2.

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
