[Supplementary Material]

## Supplementary Material

### A  Comparison with the Translation Invariant Occlusive Model

The generative model of invariant and occlusive image components can be formulated into a layered structure. An image patch $\vec{y} = (y_1, \ldots, y_{D^2})$ contains a couple of layers, and each layer is associated with a component (in total $H$ components). A component in the layer closer to the camera occludes the components in the layers further away. The layer assignment $\phi_h$ defines the layer number that the component $h$ is assigned. The larger the value of $\phi_h$ is, the closer it is from the camera, and the value "0" indicates that the component is not in the image. Therefore, occlusion for a local feature at dimension $d$ can be defined as:

$$m_d = \arg\max_h \{(T_{x_h} \vec{v}_h)_d \phi_h\} \tag{10}$$

where $m_d$ gives the index of the component responsible for generating the feature at the dimension $d$, and the binary variable $\vec{v}_h$ defines the transparency of the component $h$. The value of $\vec{v}_d$ is determined according to a prior distribution:

$$p(\vec{v}_h) = \prod_{d=1}^{D^2} \alpha_{hd}^{v_{hd}} (1 - \alpha_{hd})^{(1 - v_{hd})} \tag{11}$$

where $\alpha_{hd}$ defines the prior probability of the dimension $d$ of the component $h$ to be not transparent. Following the same definition of translation, the observation probability can be defined as:

$$p(\vec{y} \,|\, \vec{m}, \vec{x}, \Theta) = \prod_{d=1}^{D} p(y_d \,|\, m_d, \vec{x}, \Theta) \tag{12}$$

$$p(y_d \,|\, m_d = h, \vec{x}, \Theta) = \begin{cases} B(y_d) & m_d = 0, \\ \mathcal{N}(y_d \,;\, (T_{x_h} \vec{w}_h)_d, \sigma^2) & \text{otherwise.} \end{cases} \tag{13}$$

Therefore, the probability distribution of $m_d$ in the exact occlusive model is a marginal distribution over a big space: the space of $\phi_h$ (the factorial of $H$) times the space of $(T_{x_h} \vec{v}_h)_d$.

The exclusive component analysis model can be considered as an approximation to the exact occlusive model, where the probability distribution of $m_d$ is approximated by a mixture model. To make learning tractable, we applied a new variational approximation scheme (Expectation Truncation, see Sec. 3) for the posterior computation.

### B  Morphological Analysis of Receptive Fields

We pursue a better understanding of the learned receptive fields by matching them against Gabor $\mathcal{G}(x, y)$ and difference-of-Gaussians $\mathcal{D}(x, y)$ functions. For each receptive field $R_h(x, y)$, we sought the eight parameters which minimized the mean squared error between the field and the Gabor-wavelet $\mathcal{G}(x, y \,;\, \mu_x, \mu_y, \Psi, \sigma_x, \sigma_y, k_0, \tau, A)$. Where $\mu_x$ and $\mu_y$ are the center coordinates of the Gabor-wavelet, $\Psi$ is its spatial rotation, $\sigma_x$ and $\sigma_y$ parameterize the shape of the Gaussian envelope, $k_0$ is a measure of the frequency of the planar wave component, $\tau$ is its phase shift and $A$ is the overall amplitude of the Gabor-wavelet:

$$\mathcal{G}(x, y) = A \, \cos\left[x' k_0 + \tau\right] \times \mathcal{N}\left(\begin{pmatrix} x' \\ y' \end{pmatrix} ; \mu = \begin{pmatrix} 0 \\ 0 \end{pmatrix}, \pm = \begin{pmatrix} \sigma_x^2 & 0 \\ 0 & \sigma_y^2 \end{pmatrix}\right) \tag{14}$$

$$= A \, \cos\left[x' k_0 + \tau\right] \times \frac{1}{2\pi\sigma_x\sigma_y} \exp\left[-\frac{1}{2} \begin{pmatrix} x' \\ y' \end{pmatrix}^T \begin{pmatrix} \sigma_x^2 & 0 \\ 0 & \sigma_y^2 \end{pmatrix}^{-1} \begin{pmatrix} x' \\ y' \end{pmatrix}\right],$$

where $\begin{pmatrix} x' \\ y' \end{pmatrix} = \begin{pmatrix} \cos\Psi & \sin\Psi \\ -\sin\Psi & \cos\Psi \end{pmatrix} \begin{pmatrix} x - \mu_x \\ y - \mu_y \end{pmatrix}$ are the translated and rotated coordinates of the function.

Similarly, again for each receptive field $R_h(x, y)$, we sought the eight parameters of the difference-of-Gaussians kernel $\mathcal{D}(x, y \,;\, \mu_x, \mu_y, \Psi, \sigma_x, \sigma_y, \gamma, A_1, A_2)$ which minimized the squared distance to

Figure 1: The receptive fields of each latent. The Receptive Fields (**A**) were fitted with 2 dimensional DoG (**B**) and Gabor (**C**) functions. The dotted lines mark receptive fields that were not approximated very well by the fitted function. The dashed lines mark receptive fields that have a more globular structure. The solid lines mark the receptive fields the were fitted accurately by a Gabor function.

each field. $\mu_x$ and $\mu_y$ are the center coordinate of the DoG kernel, $\Psi$ its spatial rotation. $\sigma_x$ and $\sigma_y$ parameterize the shape of the inner Gaussian, $\gamma$ parameterizes the size difference between the Gaussians and $A_1$ and $A_2$ specify the amplitudes of the Gaussians:

$$\mathcal{D}(x,y) = A_1 \, \mathcal{N}\left(\begin{pmatrix} x' \\ y' \end{pmatrix}; \mu = \begin{pmatrix} 0 \\ 0 \end{pmatrix}, \pm = \begin{pmatrix} \sigma_x^2 & 0 \\ 0 & \sigma_y^2 \end{pmatrix}\right) - \mathcal{A}_\in \, \mathcal{N}\left(\begin{pmatrix} x' \\ y' \end{pmatrix}; \mu = \begin{pmatrix} 0 \\ 0 \end{pmatrix}, \pm = \begin{pmatrix} \sigma_x^2\gamma^2 & 0 \\ 0 & \sigma_y^2\gamma^2 \end{pmatrix}\right)$$

$$= \frac{A_1}{2\pi\sigma_x\sigma_y} \exp\left[-\frac{1}{2}\begin{pmatrix} x' \\ y' \end{pmatrix}^T \begin{pmatrix} \sigma_x^2 & 0 \\ 0 & \sigma_y^2 \end{pmatrix}^{-1} \begin{pmatrix} x' \\ y' \end{pmatrix}\right] - \frac{A_1}{2\pi\sigma_x\sigma_y\gamma^2} \exp\left[-\frac{1}{2}\begin{pmatrix} x' \\ y' \end{pmatrix}^T \begin{pmatrix} \sigma_x^2\gamma^2 & 0 \\ 0 & \sigma_y^2\gamma^2 \end{pmatrix}^{-1} \begin{pmatrix} x' \\ y' \end{pmatrix}\right]$$

As a guideline for the classification of a receptive field as being globular we used the aspect ratio ($\sigma_x/\sigma_y \leq 2$) of the DoG fitted to it was smaller than 2.0 . Furthermore, if the absolute error of the best matching DoG function was smaller than the absolute error of the best matching Gabor wavelet the receptive field would more likely be treated as a globular. A small difference between the errors of a match with DoG and a match with a Gabor function would result in treating the filter as *ambiguous*. The functions were fitted using a standard least-square optimization method [37]. The experimental data consisted of 100 receptive fields of $16 \times 16$ pixels. As a result 15 receptive fields were classified as globular fields, 58 Gabor-like and the remaining 27 were considered *ambiguous*.

Figure 2: The different receptive fields estimated with different $\lambda$ values.

## C   Additional details about estimated receptive fields

The experimental result in Sec. 5 shows the estimated receptive fields with a particular regularization parameter ($\lambda = 0.10$). To visualize the influence of different regularization parameters, Fig. 2 shows the estimated receptive fields with different regularization parameters. As can be seen, the estimated receptive fields are insensitive to the changes of $\lambda$.

When estimating a receptive field, the input stimuli were shifted according to the estimated position of the corresponding component (see Eqn. 8). Estimating the receptive fields shown in the main text and above used the translation matrices $\bar{T}_{x_h}$ without the cyclic boundary condition, which is different from the translation matrices in the generative. In these translation matrices, the pixels that

Figure 3: The estimated receptive fields with cyclic boundary conditions ($\lambda = 0.10$).

$\alpha$           $W$           receptive fields

Figure 4: The learned model parameters and estimated receptive fields from the image patches with ZCA-whitening. The heat maps are normalized for individual components. The receptive fields were estimated with $\lambda = 0.11$.

are moved in from outside the image patch were set to zero. As a comparison, we also estimated receptive fields with cyclic boundary condition, where the translation matrices from the generative model was used. The estimated receptive fields are shown in Fig. 3.

## D  Learning with different pre-processing

We applied ZCA-whitening to the same set of image patches as mentioned in Sec. 5 ($N = 100,000$ with a resolution $16 \times 16$), and ran our learning algorithm with the same parameter setting as in Sec. 5. The learned model parameters and the inferred receptive fields are shown in Fig. 4. As a comparison with the results in Sec. 5, this receptive fields contain more high frequency Gabors.

We also ran our learning algorithm with the same parameter setting as in Sec. 5 to the same set of image patches without whitening. The learned model parameters and the inferred receptive fields are shown in Fig. 5. Note that these simulations are some first preliminary results. We observed that without withening parameter convergence is much slower.

Figure 5: The learned model parameters and estimated receptive fields from the image patches without whitening. The heat maps are normalized for individual components. The receptive fields were estimated with $\lambda = 0.56$.