[Reviews · NeurIPS 2013]

Submitted by Assigned_Reviewer_2

This paper presents a generative model for natural image patches which takes into account occlusions and the translation invariance of features. The model consists of a set of masks and a set of features which can be translated throughout the patch. Given a set of translations for the masks and features the patch is then generated by sampling (conditionally) independent Gaussian noise. An inference framework for the parameters is proposed and is demonstrated on synthetic data with convincing results. Additionally, experiments are run on natural image patches and the method learns a set of masks and features for natural images. When combined together the resulting receptive fields look mostly like Gabors, but some of them have a globular structures.

Quality:
The model is interesting and accounts for some very important constituents of natural images. I like the explicit modeling of translation invariance and the relation drawn between this and convolutional networks in the discussion. Results are quite interesting as well.

I have several reservations though which I would be happy if the authors can address to.
My main concern is about the conditional independence assumption given the mask and features (with locations) - why was the noise chosen to be pixel-wise independent? This really limits the expressive power of the model in my opinion, as it only allows the resulting patches to have a "sprite" like structure, with similar features just masked and translated. I would be happy to see samples from the model as well, and compare them to natural image patches.
Additionally, I would love to see what happens when you train the model on non-filtered (unwhithened) patches, and see the effect of whitening here, as I suspect it has a large part of the resulting receptive fields.
Finally, the background model seems both artificial and simplistic to me. I am not sure what "background" is even in natural images, it is mostly other elements of the scene just scaled down, or blurred - why not just constrain all the pixels to be covered by at least one mask? It would have been nice if a "background" element was learned automatically from the data (flat mask with simple features, for example).

Clarity:
The paper is all in all well written, but since the model is quite complex there are many different parameters, and I must say that sometimes their definition is hidden in some inline equation which makes it harder to follow. I would suggest making Figure 1 more approachable by replacing the mask and features used to something synthetic which would convey the message. The current ones used not very intuitive (for example, if the feature and mask would be switched I don't think anyone would notice). A simple mask and a simple texture would probably be easier to understand here.

Originality:
Looks like an original work with an interesting model and good analysis.

Significance:
This work would be interesting to the natural image statistics community, as well as to parts of the neuroscience and sparse coding people around.
Summary: An interesting paper with a detailed model which accounts to some basic properties of natural images. While there are some concerns here, all in all this is good solid work.

Submitted by Assigned_Reviewer_3

210 - What Are the Invariant Occlusive Components of Image Patches? A Probabilistic Generative Approach

The paper demonstrates that learning and inference are feasible in a nonlinear generative model of natural images that captures translation invariance and occlusion. This is an interesting extension of previous work on Occlusive Component Analysis. When applied to natural image patches, it confirms the previous finding that modeling occlusions leads naturally to globular receptive fields beyond the usual oriented, Gabor-like filters.
The paper is technically sound, well written, and puts the presented work in the larger context of probabilistic models of images. The results are not surprising given previous work on OCA, but the technical advance over convolutional networks (namely the occlusive nonlinearity, and the ability to learn a substantially larger number of components) is impressive.

- Quality
The model is a translation invariant extension of OCA, that includes all possible planar translations. Therefore inference is intractable, but the paper clearly demonstrates an efficient approximation based on preselection.
The results on natural images are analyzed quantitatively, by fitting linear receptive fields to the inferred components, and showing that the majority of the RFs are oriented Gabors, but a large proportion of RFs can be characterized as globular or containing more complex structure.

Two aspects of the results on artificial data (Section 4) seem potentially worrying to me.

First, in Fig. 2C the system learns all the true components, plus one that was not used to generate the data; this extra component resembles the globular RFs that are a signature of this model, so isn't it worrying that the model 'hallucinates' one in a simple artificial dataset that does not contain any? How do artificial patches, for which this "dumpy" component is inferred to be present, look like? What proportion of globular fields would be found if the model was trained on noise inputs, or on occlusion-free natural image patches (eg textures)?

Second, on line 231 the authors state: "We assess the reliability of our learning algorithm by repeating the learning procedure with the same
configuration but different random parameter initializations. The algorithm covers all the generative components in 11 out of 20 repetitive runs." Doesn't this mean that on almost half the cases the training converges to the wrong solution?


- Clarity
The paper is clearly written, well organized, and contains all the information necessary to understand the model and the results. Here are some minor suggestions:
- Line 232: "access" should be "assess"
- Line 319: I think "W" should be "R" ??
- Reference 34 appears not to be used?


- Originality
To my knowledge, the main novelty of the paper is to extend OCA to include translation invariance. Inference in this model is intractable but the authors provide an efficient approximation using the existing technique Expectation Truncation. This also results in a technical advance over other convolutional network approaches in terms of the number of components that can be learned.


- Significance
The paper provides a demonstration that complex nonlinear generative models can be efficiently trained on natural image patches. It will be interesting to see whether the quantitative (components) and qualitative (globular RFs) improvements over existing invariant models translate into better performance at perceptual tasks.
Summary: The paper extends the Occlusive Component Analysis model to incorporate translation invariance, using a variational approximation to train the model on natural images. The results are somewhat expected and confirm previous findings of OCA, but the approach overall makes a step forward in demonstrating the feasibility of sophisticated generative model of complex signals.

Submitted by Assigned_Reviewer_7

The paper describes a new generative model of images, in which low-level features are first shifted and then combined according to a nonlinear, stochastic, masking process. The authors develop approximate inference and learning algorithms, and demonstrate results on grayscale image patches.

The paper is clearly written, well organized, and easy to follow. It introduces a combination of two previously explored ideas (translation invariance and occlusive image generation), so conceptually it is somewhat of an incremental advance, but the approximations to inference of occluding components are novel and lead to a new structure for model parameters (feature weights and mask probabilities). Although the results are very similar to previously reported feature learning algorithms, they seem promising, especially if such a model could be extended hierarchically.

My main concern is with the focus of the paper: is the goal to generate predictions and theories for biological processing, or is it to propose a new set of representations more useful for computer vision?

If the focus is on computer vision, the authors should explain why this solution to occlusion is better than other occlusive models (including max- rule for feature combinations, dead leaves models, and masked RBM by Le Roux, Heess, Shotton, Winn, 2011), and also why translation invariance makes the model more tractable than convolutional models. As it is, this paper presents another alternative to occlusive and translating models (though it unifies the two computations).

If the aim is to provide a theoretical result for neuroscience, the authors should emphasize what kind of predictions this model makes (or what it explains about observed properties of neurons in visual cortex). The prevalence of center-surround receptive fields has been noted and modeled previously. Several theories have been proposed for translation invariance in complex cells, and some models even derive this directly from objective functions like information maximization or temporal stability of the representation. If this model is to be taken seriously in the context of brain processing, specific, novel predictions or explanations should be offered, and aspects of the model that are not biologically plausible (like the complete translation invariance) should be addressed in the discussion. I recommend backing off the neuroscientific claims unless these can be strengthened sufficiently to be useful to experimentalists.



Other comments:

What is the benefit of the stochastic component assignment over choosing pixel value with a max rule, as in (Puertas et al, NIPS2010)? Also, the all-or- none activation of the features seems like a limitation of the proposed model.

Is it possible to relate the (feed-forward) operations in a convolutional neural network to performing approximate inference with expectation truncation? What exactly are the benefits of probabilistic pooling?

Why all the work to compute the "estimated Receptive Fields"? For visualizing and interpreting model parameters, the mask-feature product seems to work quite well. As a comparison to biology, the translation invariant receptive field is not very appropriate: complex cells are not "fully translation invariant" as claimed in the Discussion (so it's not a good characterization of a complex cell's behavior), and for simple-like cells, linear receptive fields are estimated using direct regression methods. If model units are to be interpreted as populations of cells, then wouldn't a convolutional network with replicated receptive fields be a better model? As an aside, new methods are being developed to characterize the features encoded by translation invariant neurons (e.g., Eickenberg, Rowekamp, Kouh, Sharpee, 2012; Vintch, Zaharia, Movshon, 2012). These might be worth citing, though there isn't much data analyzing large neural populations yet.

In the last paragraph, there is a mention of building hierarchical versions of this model. I am curious if the authors have more specific ideas of how multi-layered occlusive models can be constructed, and what kind of features they will extract from natural images. Specifically, would the layering/transparency be interpreted similarly at higher levels of the hierarchy, or would it simply add a nonlinear stochastic component to a deep model? Results presented here are not strikingly different from many other learning algorithms, so it is important to show that extensions to the model have promise.


Minor comments:

How are image patch boundaries handled during translation?

I am assuming the masks are constrained to be nonnegative, but the text does not specify.

What is the motivation for prefiltering with center-surround? It's true that this is comparable to the (linear component of the) transformation performed in LGN, but receptive fields are experimentally derived by correlating to pixel stimuli on the screen, not LGN outputs.

It would be helpful if a sentence or two in the paper listed all the approximations required to make the model tractable (expectation truncation, independent pixel occlusion).

Do you have any insight as to why all globular components have positive centers?
Summary: This is a clearly written paper describing that decsribes a somewhat incremental advance: the combination of two previously developed ideas. The results suggest that the learning algorithms can learn interesting structure, but so far the authors have only replicated features learned with other models.
Author Feedback

Author rebuttal: Rev #2

Cond. independence is the standard assumption for sparse coding. In general it reduces model complexity. For our model such a reduction is important for tractability. Furthermore, analytical derivation of update equations are facilitated.

A major point was neural consistency, DoG preprocessing is the most straight-forward model for LGN. We'll discuss results for PCA whitened patches and add results to the supplement. We'll also add results for non-whitened patches (components are less localized).

Purpose of the background model is mainly to absorb uncertainty and outliers (inspired by "robustified" in [13]); a more detailed model would be interesting, yes.



Rev #3

First:
The field localized to on pixels in Fig. 2c is an artifact, and can clearly be identified as such by looking at the corresponding appearance probability (Fig. 2c, top); the field has a different shape each run (only sometimes single pixel). The glob fields in Fig.3 are no artifacts, they have the same appearance probabilities as the Gabor fields (see Fig.3c which induces the field sorting of Fig.3f). Therefore, no worries here. We'll explicitly point this out, thanks.

For artif. data: prob to infer dumpy component is low because of low prior appearance prob. On noise (e.g., iid Gaussian) neither Gabors nor globular fields would be inferred. On textures, less localized fields can be expected with shapes highly dependent on the textures.

If we consider single-object images as occlusion free, then globular fields can still be expected as they are associated with corners and end-stopping.

Second:
In the exp. on artif. data, 9 out of 20 runs did not recover all the generative components. In this sense they fail, yes. The EM algorithm as gradient approach cannot guarantee a convergence to global optima. But the 9 runs not recovering all bars did recover most of them (usually 7 bars out of 8) - they are as such still reasonable solutions. Also bars stimuli have pronounced local optima, image patches less so; in all runs on patches we observed very similar solutions.


Rev #7

We'll address the raised points starting with probabilistic pooling (PP): PP in our model allows for maintaining alternative interpretations of a patch. Standard pooling picks one position for further processing. Our ideas for hierarchical extensions: a) Combination with standard deep architectures as additional layers (note our binary hiddens); b) definition of additional layers such that obj/obj parts exclusively determine low-layer variables (in gen. direction); "b" is more in the spirit of this paper, exclusiveness would persist throughout the hierarchy. Such a model would be complimentary to the undirected model LeRoux+ '11, we'll discuss.

Note that our model as "pooling stage" predicts a clearly different behavior from standard pooling: max-pooling is not affected by the presence of the same weaker feature at other positions, our model predicts a representation also of the position of the weaker feature: Given an ambiguous image patch, the model predicts V1 to represent the ambiguity (with potential strengthening of the weaker feature cell later). Feature encoding cells would be at the same time invariant to shifts within the scale of a small patch (fully invariant for a small patch btw - we'll clarify). Experimental setups with ambiguous stimuli are frequently used and measurements of detailed neural variability now become accessible (compare sampling hypothesis). Some predictions the model shares with complex cell models. New predictions are specific dependencies between feature and position encoding cells (compare "control units" in the literature).

Our main goal was answering the question how the components of image patches look like using a model combining (for the first time) occlusion and invariance. The resulting (mask/feature) encoding then immediately provokes the question if such a model can be consistent with neural encoding in V1. RF estimation is required for a comparison with neural RFs because (depending on the data) RF estimates could have been very different from an ad hoc multiplication of masks and features (due to explaining away etc.). The *RFs* do look similar to features of overcomplete linear models but note that the actual features (as mask/feature combination) are unlike those of any linear model. Predicted features are themselves experimentally testable. We think papers Eikenberg+ & Vintch+ are very relevant (we'll discuss).
Functionally, mask/feature encoding makes, eg, class memberships per pixel directly accessible (compare [20]). In general the model is functionally relevant; showing this will requires larger systems, eg, hierarchical -> beyond current scope. Main challenge of such hier. extensions is tractability, and this first has to be overcome non-hierarchically. Tractability is also the reason to favor stochastic assignment over max-rule or explicit occlusion. Also the all-or-none prior is motivated by this. Listing all approximations is a very good idea, we'll do.

Conv. approaches can be used as selection step but we need additional mechanisms enabling multiple winners (e.g., max to multi-max or inhibition of return). It should be possible to make conv. approaches more efficient than our approach.

Minor: have tried cyclic and non-cyclic -> similar results, here non-cyclic; non-negative; DoG biol. motivated; DoG preprocessing is linear mapping -> no significant difference to RF estim. in pixel space ([4] for details); will add;

Why pos. glob fields? Very good question. We get some neg. glob fields but pos. are much more frequent. Glob fields are associated with corners/end stopping so their positivity is presumably a consequence of the prevalence of convex object shapes and the fact that objects are usually brighter than the background - we'll discuss, thanks.